# Recurrent Subcutaneous Phaeohyphomycosis Due to *Medicopsis romeroi*: A Case Report in a Dermatomyositis Patient and Review of the Literature

**DOI:** 10.3390/microorganisms11010003

**Published:** 2022-12-20

**Authors:** Mohanad Aljundi, Sophie Brun, Mohammad Akhoundi, Morgane Didier, Roula Jabbour, Arezki Izri, Frédéric Caux, Gérôme Bohelay

**Affiliations:** 1Department of Dermatology, Avicenne Hospital, Assistance Publique des Hôpitaux de Paris (AP-HP), Hôpitaux Universitaires de Paris Seine-Saint-Denis (HUPSSD), 93000 Bobigny, France; 2Department of Parasitology-Mycology, Avicenne Hospital, Assistance Publique des Hôpitaux de Paris (AP-HP), Hôpitaux Universitaires de Paris Seine-Saint-Denis (HUPSSD), 93000 Bobigny, France; 3Inserm UMR 1125 Li2P, UFR SMBH Léonard de Vinci, Université Sorbonne Paris Nord (USPN), 93000 Bobigny, France; 4Department of Pneumology, Avicenne Hospital, Assistance Publique des Hôpitaux de Paris (AP-HP), 93000 Bobigny, France; 5Department of Pathology, Avicenne Hospital, Assistance Publique des Hôpitaux de Paris (AP-HP), 93000 Bobigny, France

**Keywords:** *Medicopsis romeroi*, coelomycetes, phaeohyphomycosis, dermatomyositis

## Abstract

*Medicopsis romeroi* phaeohyphomycosis is increasingly reported in immunocompromised patients living in or originating from tropical and subtropical areas. We report a case of subcutaneous phaeohyphomycosis caused by *M. romeroi* in a 56-year-old Malian woman residing in France for 20 years. She developed a small nodule on her dominant hand’s ring finger 15 months after starting immunosuppressive medications for paraneoplastic dermatomyositis. A first surgical debridement was followed by a local recurrence. Despite a second surgical excision combined with posaconazole treatment, the infection recurred one year after antifungal therapy discontinuation. A wide excision was performed again, and antifungal therapy was resumed and maintained for six months, resulting in the absence of relapse during the 18 months following the surgery. This case highlighted the high risk of relapse in immunocompromised patients, suggesting the need for long-term follow-up and prolonged antifungal treatment following surgical excision in cases with sustained immunosuppression. The literature review was performed according to PRISMA guidelines and included 51 scientific publications. A noteworthy predominance of the subcutaneous phaeohyphomycosis presentation was found in immunocompromised patients, whereas eumycetoma had been reported in apparently healthy individuals. A combination of complete excision with antifungal treatment seemed to confer the best outcome.

## 1. Introduction

*M. romeroi* (formerly *Pyrenochaeta romeroi*) is a saprophytic dematiaceous fungus that belongs to the order of *Pleosporales*, the class of Coelomycetes, a group of filamentous fungi that are characterized by the production of conidia inside fruiting structures called pycnidia [1,2,3]. Coelomycetes cause superficial or subcutaneous infections, including onychomycosis, subcutaneous phaeohyphomycosis (PHM), and eumycetoma [3]. They are widely distributed in soil and plants, mainly in tropical and subtropical regions of the world, and can infect humans by direct inoculation into the skin. Human cases of *M. romeroi* infections are rare and manifest by subcutaneous PHM or black grain eumycetoma [4,5,6,7,8,9,10,11,12,13,14]. Herein, we report a case of recurrent *M*. *romeroi* PHM in an immunocompromised patient originating from Mali.

*M. romeroi* infection is rare, but its incidence seems to be increasing worldwide, causing diagnostic and therapeutic issues, notably in immunocompromised patients. Due to its infrequency, we performed a systematic literature review of *M. romeroi* infection cases in humans conforming to PRISMA guidelines. We provide, through this review, in-depth information about the clinical and epidemiological aspects of *M. romeroi* infection by highlighting two main distinct clinical forms, the subcutaneous phaeohyphomycosis and the eumycetoma. Furthermore, we discuss the different therapeutic modalities outcomes and hope that our results could help establish future standard guidelines for the optimal management of this rare infection.

### Case Presentation

In 2018, a 56-year-old woman of Malian origin, living in France for 20 years, was referred to our dermatological department with a small tender subcutaneous nodule on the second phalanx of the fourth finger of the right hand, slowly growing for three months. She last traveled to Mali in 2013. She had worked as a house cleaner in Mali and France and had also worked for two months as a strawberry picker 15 years ago. She has been jobless since 2014 with no history of trauma on the affected finger.

The patient had a past medical history of right breast adenocarcinoma, treated first in 2017 by chemotherapy, modified radical mastectomy with axillary dissection, and external beam radiation therapy followed by hormonal therapy. These treatments yielded a complete remission within two years of follow-up with persistent right upper limb lymphedema. She developed paraneoplastic dermatomyositis associated with autoimmune necrotizing myopathy and severe interstitial lung disease, concomitantly with the diagnosis of breast cancer. Dermatomyositis treatment consisted of prednisone (1 mg/kg/day), intravenous immunoglobulins (2 g/kg/month), and intravenous cyclophosphamide, which was initially added to the chemotherapy regimen (six cycles), and later replaced by mycophenolate mofetil (MMF, 2 g per day). Myositis was resistant to prednisone, intravenous immunoglobulins, and mycophenolate mofetil. Consequently, rituximab was added with a total of five perfusions of 1000 mg every six months.

A skin nodule appeared 15 months after starting immunosuppressive treatment. Physical examination showed a mobile, firm skin-colored subcutaneous tender nodule of 5 mm diameter with normal overlying skin along with lymphedema of the right upper extremity. Thoracic and sinus computed tomography scans and positron emission tomography scans did not show any sign of deep infection. Leukocyte differential count, C reactive protein, and plasmatic β-D-Glucans were within normal limits. Due to the initial suspicion of a synovial cyst, surgical excision was performed. A pus-filled structure without a visible cystic wall was found during the surgical procedure. Fungal, bacterial, and mycobacterial cultures were therefore performed due to suspected opportunistic infection. The histopathological examination showed an epithelioid granuloma with necrotic areas containing many neutrophils located in the dermis and the hypodermis. Periodic acid-Schiff (PAS) and Grocott-Gomori’s methenamine silver staining revealed many filamentous and yeast-like structures (Figure 1A,B).

Bacterial and mycobacterial cultures were both negative. Olivaceous grey and floccose fungal colonies were observed after three days of incubation at 28–30 °C on Sabouraud Dextrose Agar containing antibiotics (Figure 2).

Microscopic examination of the culture revealed broad, septate, branched, and dark brown hyphae with no fructification, which made their accurate identification difficult based on morphological characteristics. The sequencing of 570 bp DNA fragment of the internal transcribed spacer 2 (ITS2) determined the identity of the mold as *M. romeroi* based on ≥99% identity with sequence assigned number LT796879 (USA) in GenBank. No complimentary treatment was held as the excision appeared complete. Six months later, a small swelling appeared under the scar, quickly followed by the development of a new adjacent nodule on the proximal phalanx of the same finger, both measuring one centimeter in diameter (Figure 3).

Magnetic resonance imaging (MRI) showed a subcutaneous thick-walled nodular structure at the second phalanx and a second bi-lobed nodule at the proximal phalanx of the fourth finger with no extension beyond the hypodermis, and no tendon or bone involvement (Figure 4).

A skin biopsy taken from one of these nodules grew rapidly on culture, and the diagnosis of *M. romeroi* PHM recurrence was confirmed by the ITS2-rDNA region sequencing. In vitro antifungal susceptibility testing was not carried out initially due to the lack of sporulation. Regarding the low potential of significant drug interactions with concomitant immunosuppressive treatments, empiric treatment was initiated with oral terbinafine (TRB) (250 mg/day). Two months later, as the lesions were still growing, TRB was replaced by posaconazole (POS) tablets (600 mg on the first day, followed by 300 mg/day). The prednisone and MMF doses were not lowered due to the risk of dermatomyositis flare-up. The surgical excision of the nodules was performed three months later after stabilizing their size by POS, which was continued for three months after the surgery to prevent a relapse.

One year after POS discontinuation, a new, slowly growing 4 × 5 mm nodule was observed near an excision site. *M. romeroi* recurrence was confirmed again by culture and ITS2 region’s sequencing. In vitro antifungal susceptibility testing using Etest^®^ strips showed low minimum inhibitory concentrations (MICs) for ISC 0.012 mg/L, voriconazole (VCZ) 0.016 mg/L, and POS 0.047 mg/L. MICs for amphotericin B (AMB) and itraconazole (ITR) were high (4 mg/L and >32 mg/L, respectively) with an elevated minimum effective concentration for caspofungin (CAS) (>32 mg/L). This second recurrence was treated by wide excision with 3 mm margins along with VCZ 200 mg twice daily for six months. Complete clinical and radiologic remission was still observed 18 months after the surgery and 12 months after the cessation of antifungal therapy.

## 2. Materials and Methods

To explore the demographics, clinical characteristics, and outcomes of *M. romeroi* infection, we performed a systematic review of the literature. The present survey relied on the PRISMA guideline (Preferred Reporting Items for Systematic Reviews and Meta-Analyses) [15]. PubMed and Google Scholar databases were searched in April 2022 for articles reporting *M. romeroi* infection in humans from 1968 using the following keywords: “*M. romeroi*” and “*P. romeroi*”. The initial search in the two databases identified 753 occurrences of articles. Duplicate articles were removed (*n* = 151). Studies reporting *M. romeroi* infection in non-human species were excluded as well as studies in languages other than English or French. Studies reporting therapeutic outcomes and/or clinical characteristics of *M. romeroi* infection were included. Articles with unrelated topics (*n* = 450) or without available clinical information (*n* = 101) were excluded. Finally, 51 articles reporting 52 cases of *M. romeroi* infections were included in the analysis of the literature review (Figure 5).

These 52 cases corresponded to 38 cases of PHM, including ours [1,2,3,4,16,17,18,19,20,21,22,23,24,25,26,27,28,29,30,31,32,33,34,35,36,37,38,39,40,41,42,43,44,45] and 14 cases of eumycetoma [5,6,7,8,9,10,11,12,13,14]. Two cases published as mycetoma by Thiyagarajan et al. and Sum et al. [20,21] were classified in this review as PHM, regarding the clinical presentation and the lack of grains and bone involvement. Los-Arcos et al. held the same attitude regarding the case reported by Thiyagarajan et al. [19]. The characteristics of the 14 cases of eumycetoma and the 38 cases of PHM cases are given in Table 1 and Table 2, respectively.

The comparison of demographics, clinical aspects, and therapeutic outcomes between eumycetoma and PHM presentations of *M. romeroi* infections is presented in Table 3.

## 3. Results

### 3.1. Epidemiological Characteristics of M. romeroi Infections

The mean age at diagnosis was 52 years (range: 24–80 years). Patients with eumycetoma were younger (mean age: 42 years) than those with PHM (mean age: 54 years). There was a male predominance in patients with *M. romeroi* infection (66%), which was more important in patients with eumycetoma (85%) than in those with PHM (59%).

Half of the patients were immigrants residing in the United States of America or Europe. Among them, 37% originated from Africa and 37% from the Indian subcontinent, while autochthonous cases represented 8% of patients [6,36]. France accounted for 59% of the cases reported from Europe.

Overall, patients originated (resident or having emigrated) from the Indian subcontinent in 50% of cases, from African countries in 27% of cases, from Southeast Asia in 14% of cases, from South America in 4%, and from Europe in 4%. Similar proportions regarding the geographic origin were found between the eumycetoma and PHM cases (Table 3).

An interval of 1–11 years (mean: 6 years) was observed between the last trip to tropical/subtropical countries and the onset of clinical symptoms in the cases reported from Europe and the USA.

A history of agricultural activity was reported in 23% of all cases, 36% of eumycetoma cases, and 18% of PHM cases.

### 3.2. Clinical Characteristics of M. romeroi Infections

All *M. romeroi* eumycetoma cases with available clinical information showed features of eumycetoma with subcutaneous swellings mostly involving the foot, draining sinuses, and the presence of black grains (Table 3). Bone involvement was reported in 83% of eumycetoma cases.

Most PHM cases were of the subcutaneous form and were described as subcutaneous swelling, nodule, or abscess. Draining sinuses without grains were found in 13% of cases, and verrucous lesions in 5% of cases. A superficial form of PHM was reported in one case as a toe wound [28].

Most patients with PHM exhibited single lesions (76%), while multiple lesions (defined as ≥3 lesions) were described in 24% of cases. All mycetoma cases consisted of single lesions. Lesions were described as painful in 53% of all cases, 60% of eumycetoma cases, and 52% of PHM cases. Lesions were slowly progressing in 77% of the PHM cases, while rapid growth was reported in 23% of patients. The most common sites of infection were the lower extremities which were affected in 75% of all cases and in 71% of those with PHM. Upper extremities were involved in 20% of all cases and 24% of PHM ones.

Extra-cutaneous involvement was reported in four PHM cases (10%). Ocular PHM was reported in two cases [22,23] and manifested as endophthalmitis. Osteoarticular involvement was described in one case of post-traumatic gonarthritis [3] (Table 2). Visceral PHM was described in a unique case of liver abscess invading the spine and the lung in a patient with chronic granulomatous disease [24].

### 3.3. Underlying Diseases

Thirty-three patients (63%) with *M. romeroi* infection were immunosuppressed. While no comorbidity was reported in *M. romeroi* eumycetoma cases, 87% of *M. romeroi* PHM cases were reported in patients with underlying diseases that increase the risk of infection (Table 3). Solid-organ transplantation (SOT), diabetes, and systemic steroid therapy for various autoimmune or inflammatory diseases were reported in 39, 34, and 24% of PHM patients, respectively. Chronic granulomatous disease was found in one case with an aggressive *M. romeroi* infection manifested by liver and lung abscesses. Besides, the two ocular PHM cases were not associated with an underlying disease.

The immunosuppression duration before PHM onset ranged from 1 month to 15 years (mean: 36 months) in patients with an immunosuppressive condition. In patients with SOT, a mean interval of 19 months (range: 1–72 months) was noted between transplantation and the onset of PHM.

### 3.4. Treatment and Outcomes

Information regarding therapeutic management was available in 44 cases (85%), including 37 PHM cases (97%) and 7 mycetoma cases (50%). Follow-up was reported in 31 cases (60%), including 26 PHM cases (68%) and 5 eumycetoma cases (36%) (Table 3). The mean follow-up duration was 23 months in PHM cases. Disease duration before treatment was significantly longer for mycetoma cases, with a mean of 94 months as opposed to 12 months for PHM ones.

### 3.5. Treatment and Outcomes of M. romeroi PHM

Among the 37 PHM cases with information regarding treatment, first-line therapies consisted of surgery alone in 11 cases (30%), antifungal therapy alone in 11 cases (30%), or a combination of surgery and antifungal therapy in 15 cases (40%). Four cases reported no follow-up after treatment; two cases with surgery alone and two cases with a combination of surgery and antifungal therapy (Table 2). Thus, information about the therapeutic outcome was available in 33 of the 38 cases (87%). The follow-up reported after treatment was longer in patients having received a combination of surgery and antifungal treatment (mean: 23 months) in comparison with those treated with surgery alone (mean: 19 months); information about follow-up duration in patients treated with antifungal treatment without surgery was only available in the single cured case (72 months). Following this first line of treatment, complete remission was reported in 7 of the 11 patients (63%) treated with surgery, in 11 of the 15 patients (73%) treated with a combination of surgery and antifungal therapies, and in 1 of the 11 patients (9%) of patients treated with antifungal therapies alone. Lesion stability after treatment was only reported in patients not having undergone surgery and accounted for 9% of patients having had antifungal therapy alone. Treatment failure was reported in 73%, 18%, and 13% of patients treated with antifungal therapy, surgery alone, and with the combination of surgery and antifungal treatment, respectively.

Twelve cases (32%) refractory to first-line therapy had a second-line treatment that consisted of surgery alone in only one case, antifungal therapy alone in eight cases, and a combination of surgery and antifungal therapy in three cases. The exclusive antifungal treatment in the second line resulted in lesion stability in three cases (37% with a mean follow-up of 14 months), a progression in four cases (50%), and complete remission in one case (12%, with a six months follow-up), while a combination of surgery and antifungal treatment resulted in a cure in two cases (67% with a mean follow-up of eight months) and stability in one case (33% with no follow-up available). The exclusive treatment by surgery was carried out only in one case with no outcome available [18] (Table 2).

When considering different lines of treatments of PHM as independent events (Table 3), surgical treatment alone resulted in complete remission in 78% (mean follow-up of 19 months) and failure (relapse) in 25%, while antifungal treatment alone led to complete remission in only 12% with a 62% failure rate (lesions progression). However, partial improvement or stability of lesions was obtained in about 25% using only the antifungal medication. The best results were obtained by combining antifungal therapy with surgery, with a complete remission in 84% (mean follow-up of 18 months) and a 16% failure rate (progression or relapse).

No association between treatment outcome and the patient’s immune status was found by comparison of the only surgery and the combined surgery with antifungal treatment groups (immunosuppression being reported among 86% and 87% of the cured patients in both groups, respectively).

Immunosuppressive therapy reduction was reported only in four cases. One of them was cured only by surgery [25], and another one by a combination of surgery and antifungal treatment [26]. One case showed a progression despite antifungal treatment [27], while no outcome was available for the last one [28].

### 3.6. Antifungal Medications in PHM Cases

Antifungal treatment duration ranged from 2 weeks to 3 years, with a median of 2 months and a mean of 3.5 months. The latter was longer in cases treated with antifungal treatments only (mean: 5.3 months) in comparison to cases treated with surgery combined with antifungal medications (mean: 2.75 months). However, the median duration of antifungal treatment was not different between the two groups (two months). ITC, VRC, POS, and LAMB were the most used antifungals. The paucity of available clinical data does not allow us to suggest which molecule could be the most effective against *M. romeroi* infection.

Antifungal susceptibility testing was carried out in 11 cases of *M. romeroi* PHM, including ours, and one case of *M. romeroi* eumycetoma (Table 4). Low minimum inhibitory concentrations (MICs) were found for POS, VCZ, and ISC in contrast to itraconazole (ITC), fluconazole (FLC), and AMB. TRB showed variable results with a mean MIC of 2.79 mg/L. High minimum effective concentrations (MECs) were observed for CAS and micafungin as opposed to anidulafungin.

### 3.7. Treatment and Outcomes of M. romeroi Eumycetoma

Treatment information was available for 7 of the 14 cases of *M. romeroi* eumycetoma, with 10 treatment events reported overall. It consisted of antifungal treatment alone in three cases, surgery alone in three cases, and a combination of surgery and antifungal treatment in one case. No complete remission was reported with either of these treatment modalities. However, antifungal treatment resulted in a partial improvement in 33% (two out of six antifungal treatment events, Table 3). Surgery alone was held in three cases, with no outcome available in two and a relapse reported in the third. A combination of surgery and antifungal treatment was reported in a single case and resulted in a relapse.

## 4. Discussion

*M. romeroi* was reported to cause an increasing number of cases of subcutaneous PHM in the last decades, while it had been reported previously to cause black grain eumycetoma [46] (Table 1 and Table 2). PMH is characterized by the presence in tissue samples of septate dark hyphae, pseudohyphae, and yeast-like structures. Chromoblastomycosis and eumycetoma, both induced by dematiaceous fungi, can be differentiated from PHM by the presence of sclerotic bodies in the first and grains in the second one [47].

Eumycetoma was reported in younger patients with no obvious comorbidities [4,5,6,7,8,9,10,11,12,13,14]. Subcutaneous PHM is, as shown in our literature review, the most frequently reported clinical presentation (65% of the literature review cases). It is responsible for nodulocystic subcutaneous lesions and abscesses affecting mostly immunocompromised patients with an immunosuppression course ranging from one month to many years [1,16,19,20,21,22,23,24,25,26,27,28,30,31,32,33,34,35,36,37,38,39,40,41,42,45]. In addition to this typical subcutaneous form of *M. romeroi* PHM, other presentations were rarely reported, including superficial, disseminated, ocular, and osteoarticular forms. Superficial skin or nail *M. romeroi* infection is exceptional as only one case manifested by a toe wound was reported [28], in addition to the isolation of *M.* romeroi from onychomycosis in the Netherlands [48]. Systemic dissemination was not associated with *M. romeroi* infection except in a single case of hepatic and lung abscesses described in a chronic granulomatous disease patient [24]. Ocular involvement has been reported in two cases [22,23]. Osteoarticular involvement is well known in *M. romeroi* eumycetoma in contrast to *M. romeroi* PHM, except for a unique case of post-traumatic septic gonarthritis without overlying skin involvement in a young man without underlying comorbidity [3].

Different factors might explain the increasing incidence of *M. romeroi* PHM cases in comparison with mycetoma ones. The wide use of immunosuppressants might favor *M. romeroi* PHM, which is considered an opportunistic infection in view of the high percentage of patients with PHM who are immunocompromised (87%). Moreover, molecular biology’s expanding employment improved species diagnosis with more accuracy. Otherwise, some previous *M. romeroi* infections could have been misdiagnosed as mycetoma, and the accurate distinction between these two phenotypes based on the presence of grains was not always applied in the literature. Garcia-Hermoso et al. found that all *M. romeroi* isolates in their study performed on Coelomycetes in Spain and France recovered from non-mycetoma subcutaneous infections [3].

The literature review revealed a long duration of immunosuppression and a long delay between the last trip to tropical/subtropical countries and the onset of *M. romeroi* PHM symptoms. This could be explained by a long-term carriage after a trauma-induced inoculation with further reactivation of *M. romeroi* [1].

There are currently no standard guidelines for the treatment of *M. romeroi* PHM, but the most effective approach seems to be the combination of complete excision and antifungal therapy, considering a reported complete remission rate of 73% in the first-line treatment and an estimated complete remission rate of at least 84% when considering different lines of treatment as independent events (Table 3). Antifungal therapy alone without excision would not be the appropriate first-line treatment because most *M. romeroi* PHM lesions could be removed completely with a simple excision, while only a 12% cure rate is reported with antifungal therapy alone [2]. Simple excision without antifungal treatment remains a reasonable option for lesions of limited size, a complete remission being reported after a first-line surgery alone treatment in 63% of cases and in 78% of cases when different lines were taken as independent events. Nevertheless, despite the fact that removal of the lesion is obtained with surgery, a complete and lasting remission could not be ascertained by the literature review as relapse might have occurred with a longer follow-up duration. Notably, like others, our case highlighted the risk of recurrence in immunocompromised patients after surgery alone in less than one year. Besides, in our case, as in others in the literature, the combination therapy did not prevent the relapse at the 12-month follow-up that occurred after apparently successful treatment. Thus, the recurrence rate could be underestimated in the literature as a lot of reported cases had a follow-up duration of less than one year. In such patients with sustained immunosuppression, the surgery combined with long-term antifungal treatment could reduce this risk. Moreover, an MRI of the involved area before surgery could be helpful in better determining the extension of the infection in the host tissue, so it probably reduces short-term relapses after surgery.

In patients under immunosuppressants, the scarcity of literature data did not allow estimation of the importance of immunosuppressive regimen reduction in the management of *M. romeroi* PHM [25,26,27,28]. However, it would be reasonable to suppose that such a reduction could have a role in minimizing the recurrence risk.

The management of *M. romeroi* eumycetoma is challenging as surgical excision is often mutilating or impossible. Our review showed the failure of *M. romeroi* mycetoma treatment with KTC, POS, and AMB in three of the 6 cases that reported treatment outcomes [9,13,14], a recurrence after surgical excision with or without antifungal treatment in 2 cases [4,14], and a slight improvement after ITC treatment in one case [9]. An exception to these poor results is one case that described great improvement after KTC treatment [12].

Regarding the limited number of reported cases, there is currently no recommendation on which antifungal therapy to use as a first-line treatment in *M. romeroi* PHM or eumycetoma. In addition, due to the frequent absence of fructification during fungal culture, there is a lack of systematic in vitro antifungal susceptibility testing that could be helpful in choosing the appropriate treatment based on MICs values. The literature showed variable results of in vitro antifungal susceptibility tests for *M. romeroi* strains. POS, VRC, and ISC showed relatively low MICs contrary to ITC and AMB (Table 4). However, no defined cutoff value exists regarding MICs of antifungal therapies in *M. romeroi* infections, and the correlation between MIC values and clinical outcome is not clear. In the study carried out by Garcia-Hermoso et al., MICs of different antifungals were tested for 46 isolates of Coelomycetes fungi recovered from clinical specimens in France and Spain [3]. MIC values of ITC and MECs of CAS were the highest, as in our literature review. TRB had low MICs, but interestingly not for some *M. romeroi* isolates [3].

## 5. Conclusions

*M. romeroi* infection should be suspected in all immunocompromised patients having subcutaneous nodules or abscesses, notably involving the extremities. Complete excision, whenever possible, is a suitable option as a first-line treatment to obtain remission. Its association with antifungal therapy might further enhance the chance of having a lasting complete remission in immunosuppressed patients, notably in areas where a second surgical procedure to treat a relapse might expose functional consequences. Reports with longer follow-up duration are required to confirm the therapeutic protocol that would offer the best outcome regarding relapse and long-lasting complete remission. The effect of immunosuppression reduction and the duration of antifungal therapy to prevent local relapses need to be clarified.

## Figures and Tables

**Figure 1 microorganisms-11-00003-f001:**
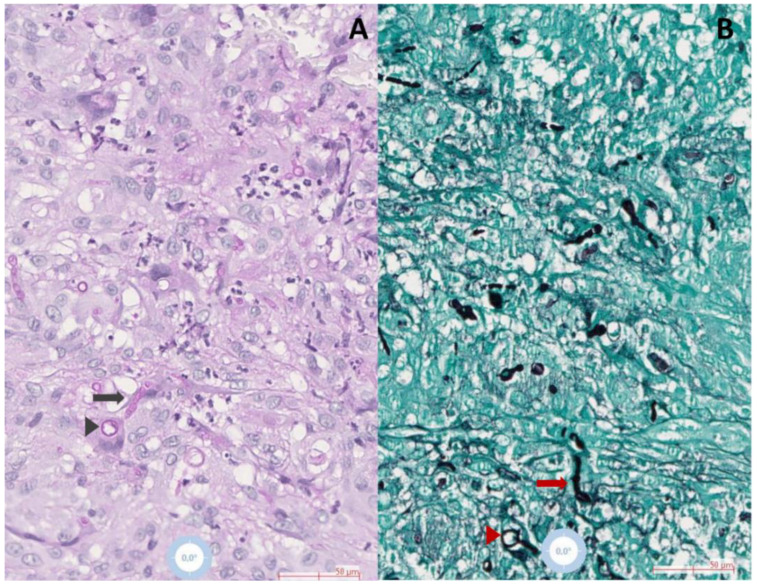
Histopathological examination of skin biopsies demonstrating hyphal (arrows) and yeast structures (arrowheads) surrounded by epithelioid, multinuclear giant histiocytes and neutrophils in (**A**) Periodic acid–Schiff (black arrow and arrowhead), and (**B**) Grocott-Gomori (red arrow and arrowhead) stained slides, observed in 20× magnification.

**Figure 2 microorganisms-11-00003-f002:**
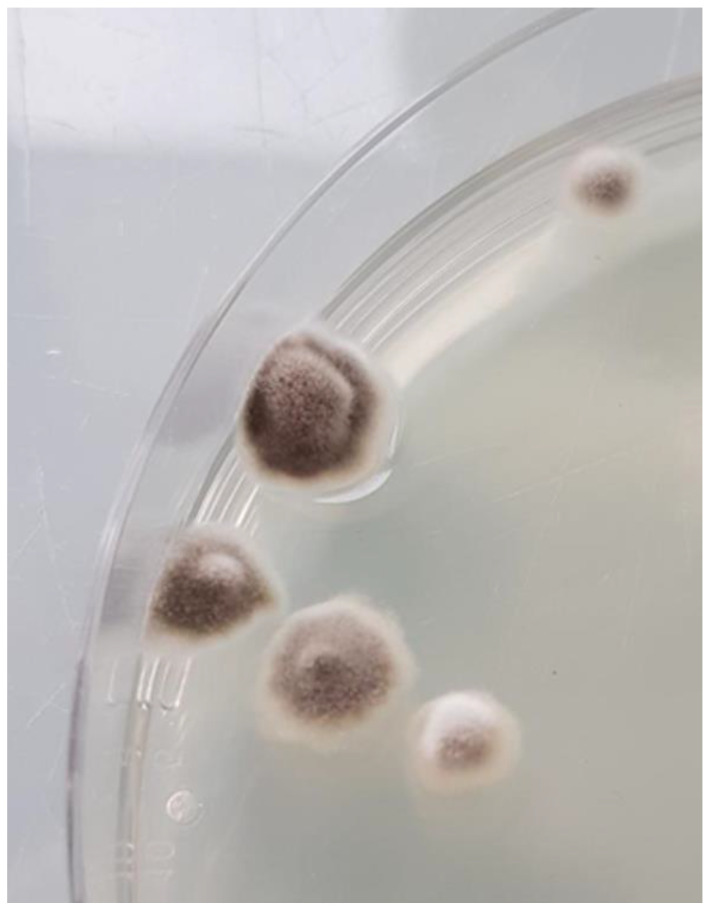
Olivaceous grey and floccose colonies of *M. romeroi* after three days of incubation at 28 °C on Sabouraud Dextrose Agar.

**Figure 3 microorganisms-11-00003-f003:**
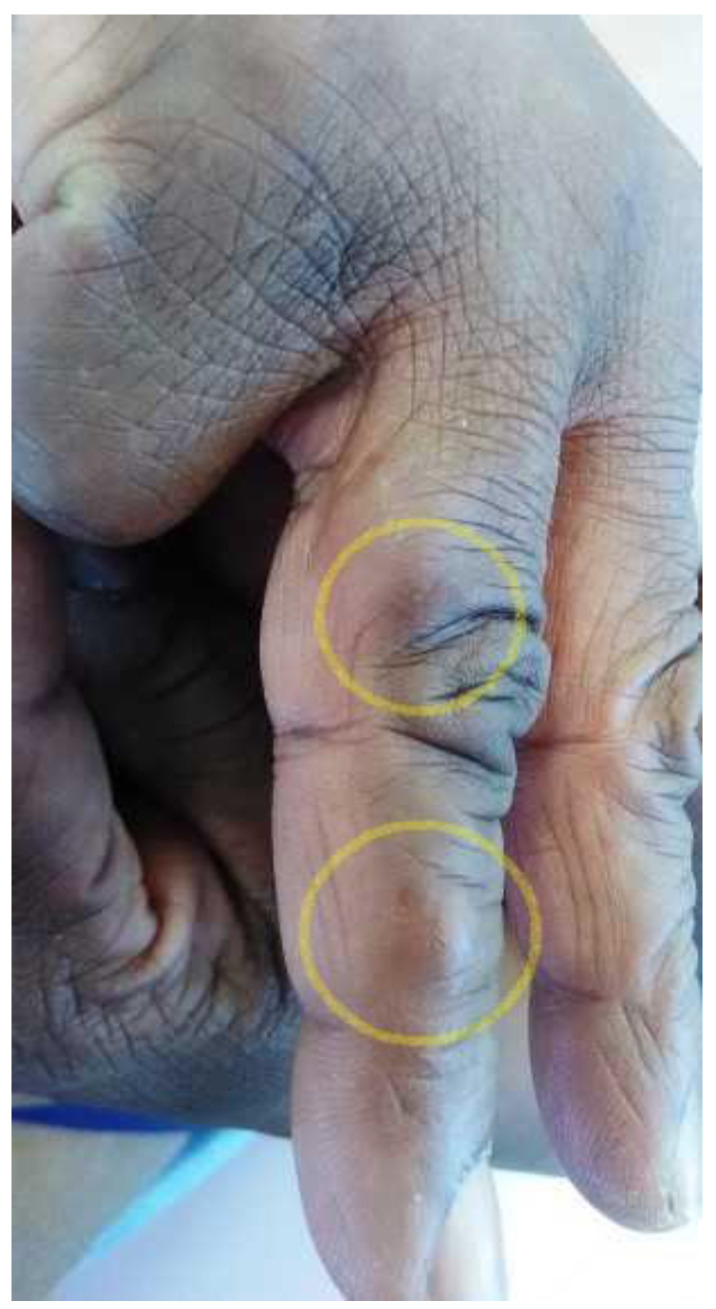
Two subcutaneous nodules on the first and second phalanges of the fourth right finger (yellow circles).

**Figure 4 microorganisms-11-00003-f004:**
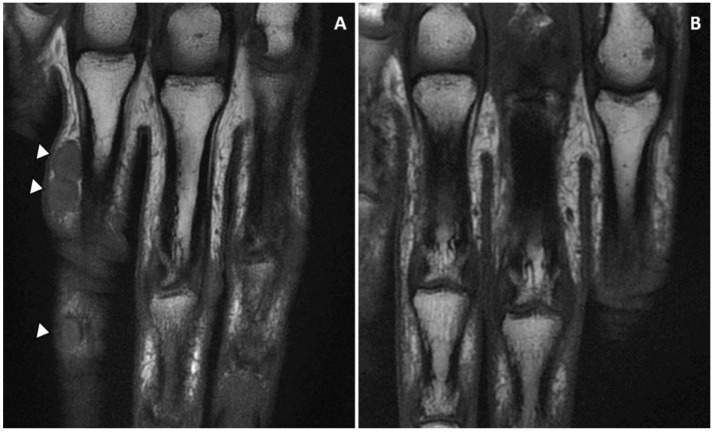
(**A**) The nodules on the first (bi-lobed) and the second phalanx of the fourth finger (right hand) were revealed by MRI imaging. The lesions were limited to the subcutaneous tissue with no tendon and bone involvement. (**B**) MRI imaging confirmed the complete remission at six months follow-up.

**Figure 5 microorganisms-11-00003-f005:**
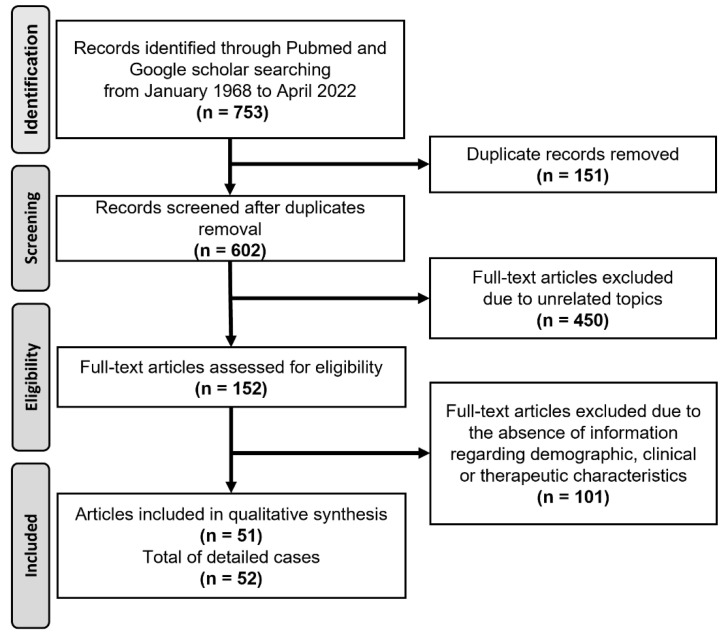
Flow chart of the literature review.

**Table 1 microorganisms-11-00003-t001:** Demographics, clinical characteristics, and treatment outcomes of the patients with *M. romeroi* eumycetoma cases in the literature.

Demographics				Clinical Characteristics				Diagnosis		Treatment	Outcome	Reference
Age (Years)	Sex	Country of Occurrence	Country of Origin	Site	Pain	Bone Involvement	Pre-Treatment Period (Months)	Histology & Culture	Molecular Identification			
63	F	Somalia	-	*NA*	*NA*	*NA*	18	Yes	No	Surgery	NA	[11] *
39	M	Cambodia	-	Foot	Yes	Yes	21	Yes	No	Excision	Recurrence	[4]
NA	M	Senegal	-	Leg	NA	NA	NA	Yes	No	NA	NA	[5]
NA	M			Sole								
NA	M			Hand								
53	M	France	-	Foot	Yes	Yes	24	Yes	No	Amputation	NA	[6]
NA	NA	Venezuela	-	NA	NA	NA	NA	Yes	No	NA	NA	[7]
37	M	India	India	Foot, ankle	No	NS	216	Yes	No	NA	NA	[8]
42	M			Foot	No	Yes	60	Yes	No			
30	F	NA	NA	Foot, leg	NA	Yes	180	Yes	No	KTC (400 mg/d) for 8 months	Great improvement without further recorded follow-up	[12] **
42	M	Brazil	-	Sole	NA	No	36	Yes	No	KTC	Failure	[9]
										ITC	Slight improvement	
36	M	India	-	Foot	NA	NA	NA	Yes	No	NA	NA	[10]
56	M	UK	Pakistan	Foot	Yes	Yes	204	Yes	Yes	ITC 200 mg b.i.d + 5FC 1000 mg t.i.d for 9 months	Failure	[13]
										VRC 200 mg b.i.d for 7 weeks, then 150 mg b.i.d for 4 years	Minimal improvement	
										POS 400 mg t.i.d for 17 months	Decrease in pain and swelling, then relapse	
24	M	India	-	NA	NA	NA	NA	Yes	Yes	AMB and surgery	Recurrence	[14] ***

Abbreviations: AMB: Amphotericin B, b.i.d: twice a day, d: day, F: female, ITC: Itraconazole, 5-FC: Flucytosine, KTC: ketoconazole, M: male, NA: not available, POS: Posaconazole, t.i.d: three times a day, VRC: Voriconazole, we: week, yr: year. ** Five cases of M. romeroi mycetoma were identified in this survey on a histological basis only. Cultures were positive for M. romeroi in only one case, so the other four cases were excluded. ** A case series of 8 patients from India, Saudi Arabia, and Yemen, the origin of the patient affected by M. romeroi was not specified. *** Case series from India, the exact year of diagnosis of individual cases, was not provided.*

**Table 2 microorganisms-11-00003-t002:** Epidemiological, clinical characteristics, and treatment outcomes of the patients with *M. romeroi* phaeohyphomycosis in the literature.

Demographics				Clinical Characteristics				Treatment							
Age (Years) /Sex	Country of Occurrence	Country of Origin *	Last Travel (Years) **	Site	Clinical Presentation	IS Duration (mo) ‡	Comorbidities	1st Line	Outcome	2nd Line	Outcome	3rd Line	Outcome	Follow-Up (mo)	Reference
45/M	France	Senegal	6	Leg	Subcutaneous swelling, sinuses	20	IM steroids, multibacillary leprosy	Surgery ITC (4 mo)	Cured	-	-	-	-	12	[31]
54/M	France	Ghana	NA	Hallux	Subcutaneous swelling	36	SOT, diabetes	VRC (6 mo)	Failure	VRC + CAS	Failure	Surgery	NA	NA	[21]
45/F	India	NA	NA	Forearm	Subcutaneous swelling/Verrucous plaque	0	None	Surgery	Cured	-	-	-	-	12	[29]
43/M	UK	NA	NA	Arms Legs	Nodules	12	SOT	LAMB	Stability	ITC (0.5 mth)	Stability	INF-γ (1.5 mth) IST withdrawal	Cured	20	[16]
47/F	Kuwait	India	3	Finger	Subcutaneous swelling	34	ALL Chemotherapy	Surgery	Cured	-	-	-	-	2	[32]
57/M	UK	Bangladesh	8	Knee	Subcutaneous swelling, sinuses	8	SOT	Surgery VRC (2 mths)	Cured	-	-	-	-	6	[20]
66/M	France	Central Africa	10	Heel	Verrucous nodule	14	SOT	Surgery IST reduction	Cured	-	-	-	-	9	[25]
78/M	Taiwan	NA	NA	Forearm	Papulopustular lesions	72	OS	Surgery ITC (6 mths)	Failure	LAMB (0.75 mth)	Cured	-	-	6	[33]
55/M	Singapore	NA	NA	Thigh	Nodule	12	SOT	Surgery	Failure	ITC (36 mths)	Stability	-	-	36	[34]
25/F	India	NA	NA	Eye	Red painful eye	0	None	FLU (4 wks)	Failure	Topical VRC + Oral KTC (0.5 mth)	Failure	Surgery Intravitreal VRC Intravitreal AMB	Cured	6	[22]
50/F	India	NA	NA	Foot	Subcutaneous swelling	12	Diabetes	Surgery ITC (0.5 mth)	Cured	-	-	-	-	3	[35]
63/M	USA	NA	NA	Knee	Nodular plaque & sinuses	NA	SOT	Surgery VRC (1 mth)	Failure	Surgery	NA	-	-	NA	[18]
27/F	Belgium	Gambia	NA	Hallux	Wound	12	SOT	Topical TRB (9 mths) IST reduction	NA	-	-	-	-	NA	[28]
61/F	India	NA	NS	Index	Subcutaneous swelling	120	OS and MTX for RA	Surgery ITC (3 mths) IST reduction	Cured	-	-	-	-	6	[26]
88/M	UK	NA	NA	Hand	Nodule	NS	OS for sarcoidosis, LAD, COPD	Surgery	Cured	-	-	-	-	0.33	[36]
47/F	France	Benin	NA	Foot	Subcutaneous swelling	12	Diabetes	Surgery	Cured	-	-	-	-	24	[37]
59/F	France	Sri Lanka	NA	Foot	Subcutaneous swelling	NA	Diabetes, OS for polymyalgia rheumatica	Surgery	Cured	-	-	-	-	96	[2]
73/F	France	India	NA	Foot and leg	Subcutaneous swelling	NA	OS for Giant cell arteritis	Surgery VRC (0.75 mth)	Cured	-	-	-	-	84	
65/M	France	West Africa	NA	Knee	Abscess	NA	SOT	POS (1 mth)	Cured	-	-	-	-	72	
53/M	France	Pakistan	NA	Foot	Abscess	NA	SOT	POS (0.5 mth)	Failure	Surgery LAMB (2 mths)	Cured	-	-	10	
43/M	India	NA	NA	Thigh, leg, toe	Nodules, sinuses	6	SOT Diabetes	TRB + ITC (2 mths) IST reduction	Failure	Surgery VRC	Stability	-	-	NA	[27]
48/M	India	NA	NA	Foot	Nodule	3	Diabetes, lepromatous leprosy	Surgery ITC	NA	-	-	-	-	NA	[38]
68/F	UK	Nepal	NA	Foot, toes	Multiple nodules	NA	SOT Diabetes	Surgery LAMB (1 mth) then VRC (4–12 mths)	Cured	-	-	-	-	NA	[43]
18/M	USA	NA	NA	Liver and lung	Abscess	NA	CGD	NA	Died	-	-	-	-	NA	[24]
65/M	Spain	India	1	Foot	Nodule	1	SOT, Diabetes	Surgery POS (1 mth)	Cured	-	-	-	-	48	[19]
56/F	Spain	Pakistan	4	Index	Abscess	24	SOT, Diabetes	Surgery VRC (6 mths)	Cured	-	-	-	-	36	
80/M	Thailand	NA	NA	Foot	Nodule	48	Diabetes	Surgery Glycemic control	Cured	-	-	-	-	9	[39]
65/F	USA	Philippines	NA	Foot	Subcutaneous swelling	72	SOT	Surgery VRC then POS (3 mths)	Cured	-	-	-	-	24	[1]
33/M	France	Bangladesh	NA †	Knee	Monoarthritis	NA	None	Surgery POS	NA	-	-	-	-	NA	[3]
NA/M	France	Senegal	NS	Knee	Subcutaneous swelling	NA	Diabetes, CRF	Surgery	NA	-	-	-	-	NA	
27/F	India	NA	NA	Buttock	Nodule	NA	IM steroids for ITP	Surgery	NA	-	-	-	-	NA	[40]
30/M	France	Guinea	9	Hallux	Nodule	15	SOT	VRC (2 mths)	Failure	Surgery VRC (0.5 mth)	Cured	-	-	6	[41]
60/M	India	NA	NA	Legs, buttocks	Nodules, sinuses	12	OS for RA	AMBD + CAS	Failure	ITC	Failure (death)	-	-	NA	[30]
78/F	Taiwan	NA	NA	Arm	Subcutaneous nodules	180	OS for RA	ITC (2 mths)	Failure	-	-	-	-	NA	[42]
40/M	India	NA	NA	Foot	Subcutaneous swelling	78	Diabetes	Surgery ITC (1.5 mths)	Cured	-	-	-	-	3	[45]
62/M	India	NA	NA	Foot	Subcutaneous swelling	0	None	Surgery ITC (0.75 mths)	Cured	-	-	-	-	6	[44]
64/M	USA	Laos	NA	Eye	Painful red eye	0	None	ITC (4 mths) Topical VRC Intravitreal AMB	Failure	VRC	Failure	Surgery VRC	Cured	3	[23]
56/F	France	Mali	4	Finger	Subcutaneous nodule	15	OS, RTX, MMF for Dermatomyositis Diabetes	Surgery	Relapse	TRB, POS (5 mths)	Stability	Surgery POS (3 mo)	Relapse	12	Current case

Abbreviations: AMBD: Amphotericin B deoxycholate, ALL: acute lymphoblastic leukemia, CAS: Caspofungin, CGD: chronic granulomatous disease, COPD: chronic obstructive pulmonary disease, d: day, EVE: Everolimus, F: female, IST immunosuppressive therapy, INF-γ: interferon gamma, IM: intramuscular, ITP: idiopathic thrombocytopenic purpura, IVIG: intravenous immunoglobulins, ITC: Itraconazole, LAD: Linear IgA dermatosis, LAMB: liposomal Amphotericin B, M: male, mth: month, MMF: mycophenolate mofetil, MTX: Methotrexate, NA: not available, OS: oral steroids, POS: Posaconazole, RA: rheumatoid arthritis, RTX: Rituximab, SOT: solid organ transplantation, Tac: Tacrolimus, TRB: Terbinafine, Tx: treatment, VRC: Voriconazole, yr: year, we: week. * Origin for migrants; ** Duration (yr) between last travel to tropical/subtropical country and the onset of disease; ‡ before the onset of symptoms; † The patient already had a knee abscess when he came to France.

**Table 3 microorganisms-11-00003-t003:** Demographics, clinical characteristics, and outcome in patients with phaeohyphomycosis and eumycetoma caused by *M. romeroi*.

Demographics	All Cases (*n* = 52)	Eumycetoma (*n* = 14)	Phaeohyphomycosis (*n* = 38)
Male gender, *n* (%) *	34 (66%)	11 (85%)	23 (60%)
Age at diagnosis (yr), mean, (range)	52 (18–88)	42 (24–63)	54 (18–88)
Geographical region of origin, *n* (%) **			
Indian subcontinent, *n* (%)	24 (51%)	5 (42%)	19 (54%)
Africa, *n* (%)	13 (27%)	4 (33%)	9 (26%)
Southeast Asia, *n* (%)	7 (15%)	1 (8%)	6 (17%)
Europe, *n* (%)	2 (4%)	1 (8%)	1 (3%)
South America *n* (%)	1 (2%)	1 (8%)	0
Patients resident in Europe or USA, *n* (%) *	25 (49%)	2 (15%)	23 (60%)
Tropical/subtropical origin in Europe/USA cases *n* (%) ‡	19 (90%)	1 (50%)	18 (95%)
Time delay between last travel and onset of disease (yr), mean	6	11	5
Agricultural work, *n* (%)	12 (23%)	5 (36%)	7 (18%)
Clinical characteristics			
Single lesion, *n* (%) ⁋	40 (82%)	11 (100%)	29 (76%)
Multiple lesions, *n* (%)	9 (18%)	0	9 (24%)
Painful lesion (%) ¥	16 (53%)	3 (60%)	13 (52%)
Growth speed §			
Slow (%)	10 (77%)	NA	10 (77%)
Fast (%)	3 (23%)	NA	3 (23%)
Presentation			
Nodules (%)	33 (63%)	NA	33 (87%)
Verrucous lesions (%)	2 (4%)	NA	2 (5%)
Draining sinus (%)	-	-	5 (13%)
Body area involved Þ			
Lower limb, *n* (%)	37 (75%)	10 (91%)	27 (71%)
Upper limb, *n* (%)	10 (20%)	1 (9%)	9 (24%)
Underlying disease, *n* (%)	32 (61%)	0	32 (84%)
Solid-organ transplantation	15 (29%)	0	15 (39%)
Systemic steroids for inflammatory diseases	9 (17%)	0	9 (24%)
Diabetes	13 (25%)	0	13 (34%)
None Disease mean duration before treatment (mth), (range)	19 (36%) 34 (1–204)	14 (100%) 94 (18–204)	5 (13%) 12 (1–48)
Outcome after treatment			
First-line treatment outcome, *n* of cases	44	7	37
Surgery alone	14	3	11
Complete remission *n* (%)	7 (50%)	0 (0%)	7 (63%)
Antifungal treatment alone	14	3	11
Complete remission *n* (%) Stability/partial improvement	1 (7%) 2 (14%)	0 (0%) 1 (33%)	1 (9%) 1 (9%)
Surgery and antifungal treatment	16	1	15
Complete remission *n* (%)	11 (68%)	0 (0%)	11 (73%)
*n* of treatment events considering all lines received as independent events	61	10	51
Surgery alone ŧ	15	3	12
Complete remission *n* (%)	7 (58%)	0 (0%)	7 (78%)
Antifungal treatment alone £	24	6	18
Complete remission *n* (%)	2 (9%)	0 (0%)	2 (12%)
Stability/partial improvement	6 (27%)	2 (33%)	4 (25%)
Surgery and antifungal treatment ©	22	1	21
Complete remission *n* (%)	16 (80%)	0 (0%)	16 (84%)

Abbreviations: mth: month, NA: not available. ***** Based on 51 cases (sex and country of residency not available in one mycetoma case), ****** based on 50 cases (origin not available in three and two cases of PHM and mycetoma, respectively), **‡** based on 21 cases (origin was not defined in four cases reported from Europe and USA), ⁋ based on 49 cases (clinical information not available in three mycetoma cases), ¥ based on 30 cases (information about pain was available in 5 and 25 cases of mycetoma and PHM respectively, § based on 13 cases (information about growth speed was only available in 13 cases), Þ based on 49 cases (information about the site was not available in three mycetoma cases). ŧ based on 12 events (information about surgery alone outcome was not available in three mycetoma cases), £ based on 22 events (information about the outcome was not available in two PHM cases, © based on 20 events (information about the outcome was not available in two PHM cases.

**Table 4 microorganisms-11-00003-t004:** Minimum inhibitory concentrations (MIC) or effective concentrations (MEC) of *M. romeroi* in mg/L for different antifungals *.

	KTC (MIC)	TRB (MIC)	FLC (MIC)	ISC (MIC)	ITC (MIC)	POS (MIC)	VRC (MIC)	CAS (MEC)	ANF (MEC)	MFG (MEC)	AMB (MIC)
PV. Venugopal et al., 1993 [12]	0.5	-	-	-	-	-	-	-	-	-	-
Cerar et al., 2009 [13]	-	-	-	-	-	0.25 MFC 1.0	>8 MFC > 8	-	-	-	-
Badali et al., 2010 [29]	-	-	>64	0.125	0.5	0.5	4.0	8.0	1.0	-	4.0
Khan et al., 2011 [32]	-	-	>256	-	3.0	0.064	0.008	6.0	0.5	-	8.0
Abdolrasouli et al., 2016 [36]	-	0.125	>64	-	0.5	0.25	0.5	4.0	-	-	0.25
Guégan et al., 2016 [2]	-	0.06	-	-	4	2	0.125	1	-	4	0.25
	-	0.25	-	-	8	4	0.5	1	-	2	0.5
	-	0.06	-	-	4	8	0.5	0.25	-	8	1
	-	2.79	128	-	5.77	1.80	1.81	4.6	-	4.6	2.78
Los-Arcos et al., 2019 [19]	-	0.25	-	-	8	0.5	2.0	1.0	-	-	0.25
	-	>16	-	-	>8	2.0	0.5	>16	-	-	8.0
Lieberman et al., 2019 [1]	-	-	-	-	>16	0.5	2.0	-	-	-	-
Current case	-	-	-	0.012	>32	0.047	0.016	>32	-	-	4
Median	0.5	0.25	96	0.0685	5.77	0.5	0.5	4.3	0.75	4.3	1.89

Abbreviations: AMB: Amphotericin B, ANF: Anidulafungin, CAS: Caspofungin, FLC: Fluconazole, ISC: Isavuconazole, ITC: Itraconazole, KTC: Ketoconazole, MEC: minimum effective concentration, MFC: minimum fungicidal concentration, MFG: Micafungin, POS: Posaconazole, TRB: Terbinafine, VRC: Voriconazole, -: in vitro susceptibility test not performed for this molecule. * The MIC is recorded for azoles and terbinafine, whereas the MEC is reserved for the echinocandins–caspofungin, micafungin, and anidulafungin. MFC refers to the minimum concentration that reduces the viability of the fungal inoculation by ≥99.9%.

## Data Availability

Not applicable.

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
