# Peer review of "Recurrent Subcutaneous Phaeohyphomycosis Due to Medicopsis romeroi: A Case Report in a Dermatomyositis Patient and Review of the Literature"

_microorganisms, 2022, doi:10.3390/microorganisms11010003_

Round 1

Reviewer 1 Report

The manuscript by Aljundi et al. describes a case of phaeohyphomycosis caused by Medicopsis romeroi. It also provides a review of the literature of human infections caused by this rare fungal pathogen. The study is well-conducted, provides useful information and is clearly presented. Some minor revision should however be considered before publication:

-       Los-Arcos et al. 2019 (cited in the manuscript) already performed a review of the literature on M. romeroi. Was it only on SOT patients? If not, what is the added-value of the current review? Only the provision of the new cases since 2019? Or were other meta-analyses performed? Previous reviews were also performed in 2016 (Sharma et al.) and 2013 (Hsiao et al.). How is it important to perform such review every 3 years? This should be clarified and stated in the introduction.

-       The genus name Medicopsis (and Pyrenochaeta) must be written in italic, throughout the text. Also when abbreviated (“M.”).

-       Line 100: 570 bp (not pb).

-       Please provide the DNA sequence obtained during the analyses, preferably through deposit in a data repository like GenBank.

-       Line 123: replace “sensitivity” by “susceptibility”.

-       Line 134: “but for isavuconazole (ISC)” should be deleted, I believe.

-       Table 3 : in the first column (Demographics), indicate that the range is provided between brackets for “age” and “disease mean duration”.

-       Lines 216-218: the disseminated case and the 2 ocular cases are evoked but only properly described later in the text (lines 239-242). I would suggest to invert the paragraph “Underlying diseases” and “Clinical characteristics”.

-       Line 275: if there is only one case, “mean” should be deleted.

-       Line 291: “reduction of immunosuppression” is mentioned, while in table 2, it is “immunosuppressive therapy (IST) reduction”. Please be consistent. Moreover, the abbreviation “RI” should not be mentioned, since it is not used elsewhere in the text.

-       Lines 296-300: I would also mention the median treatment duration, not only the mean duration.

-       I think that the correct abbreviation for anidulafungin is ANF, not ANI.

-       Line 404: voriconazole should not be considered as a “new” drug anymore.

Author Response

Dear Editor

We sincerely thank you and kind reviewers for their thoughtful comments and constructive suggestions which helped improve the manuscript. Please find below a detailed point-by-point response to the comments raised by reviewers. The responses have been highlighted in bold. Moreover, the modified sentences in the text have been indicated by yellow.

Best regards

Mohanad Aljundi

- Los-Arcos et al. 2019 (cited in the manuscript) already performed a review of the literature on M. romeroi. Was it only on SOT patients? If not, what is the added-value of the current review? Only the provision of the new cases since 2019? Or were other meta-analyses performed? Previous reviews were also performed in 2016 (Sharma et al.) and 2013 (Hsiao et al.). How is it important to perform such review every 3 years? This should be clarified and stated in the introduction.

Los-Arcos et al review included only SOT patients.

Sharma et al review included only 20 cases, among them were infections caused by Pyrenochaeta species other than M. Romeroi (P. unguis-hominis, P. mackinnonii, P. keratinophilia).

Hsiao et al review was not comprehensive and furthermore, included 2 cases of infection by Nocardia brasiliensis and one by Madurella grisea (Pérez-Blanco et al).

We think that the current systematic review is the first to provide in-depth information about the clinical aspects of M. romeroi infection by distinguishing Phaeohyphomycosis from eumycetoma forms, highlighting the demographic and patients characteristics differences between mentioned two groups, and accordingly, discussing the different therapeutic modalities by further stratifying treatments into different lines.

- The genus name Medicopsis (and Pyrenochaeta) must be written in italic, throughout the text. Also when abbreviated (“M.”).

All words including Medicopsis, Pyrenochaeta, and their abbreviations were written in italic throughout the text.

- Line 100: 570 bp (not pb).

It was corrected.

-  Please provide the DNA sequence obtained during the analyses, preferably through deposit in a data repository like GenBank.

With thanks for this suggestion, we already deposited DNA sequence of our patient’s sample in GenBank and received the access number of our sequence that will be released according to their time schedule. However, the text and fasta files of our sequence were added as supplementary information (please see attachment).

- Line 123: replace “sensitivity” by “susceptibility”.

It was replaced.

- Line 134: “but for isavuconazole (ISC)” should be deleted, I believe.

It was deleted.

- Table 3 : in the first column (Demographics), indicate that the range is provided between brackets for “age” and “disease mean duration”.

The range was indicated in the first column of Table 3 for “age” and “disease mean duration”

- Lines 216-218: the disseminated case and the 2 ocular cases are evoked but only properly described later in the text (lines 239-242). I would suggest to invert the paragraph “Underlying diseases” and “Clinical characteristics”.

The two paragraphs “Underlying diseases” and “Clinical characteristics of M. romeroi infections) were inverted in the text.

- Line 275: if there is only one case, “mean” should be deleted.

It was corrected.

- Line 291: “reduction of immunosuppression” is mentioned, while in table 2, it is “immunosuppressive therapy (IST) reduction”. Please be consistent. Moreover, the abbreviation “RI” should not be mentioned, since it is not used elsewhere in the text

The phrase “reduction of immunosuppression” was replaced by “immunosuppressive therapy reduction”.

- Lines 296-300: I would also mention the median treatment duration, not only the mean duration.

We added the median values to the text as suggested (lines 322 and 325). However, there was no significant difference between the group treated with antifungal treatment only and the one treated by a combination of antifungal treatment and surgery regarding the median duration of antifungal treatment.

- I think that the correct abbreviation for anidulafungin is ANF, not ANI.

It was corrected.

- Line 404: voriconazole should not be considered as a “new” drug anymore

It was modified.

Reviewer 2 Report

Medicopsis romeroi:  all names must be in italics throughout the text

Pyrenochaeta: must be in italics

Pleosporales: must be in italics

Could you please provide the ITS region sequencing?

Figure 3: I would think it would be more appropriate to identify the area with yellow or other colored circles.

The words that make up the phrase In vitro should be in italics.

Table 1. I would suggest that the references go in the last column and it is not necessary to include the author's name or the year, since the reference number is more than enough.

References must be prepared in accordance with MDPI's specifications

Also make all changes requested in change control within the paper

Author Response

Dear Editor

We sincerely thank you and kind reviewers for their thoughtful comments and constructive suggestions which helped improve the manuscript. Please find below a detailed point-by-point response to the comments raised by reviewer 2. The responses have been highlighted in bold. Moreover, the modified sentences in the text have been indicated by yellow.

Best regards

Mohanad Aljundi

Reviewer 2

- Medicopsis romeroi:  all names must be in italics throughout the text

The corrections were done for all words including Medicopsis romeroi, Pyrenochaeta romeroi and Pleospporales.

- Pyrenochaeta: must be in italics

The corrections were done for all words including Medicopsis romeroi, Pyrenochaeta romeroi and Pleospporales.

- Pleosporales: must be in italics

The corrections were done for all words including Medicopsis romeroi, Pyrenochaeta romeroi and Pleospporales.

- Could you please provide the ITS region sequencing?

With thanks for this suggestion, we already deposited DNA sequence of our patient’s sample in GenBank and received the access number of our sequence that will be released according to their time schedule. However, the text and fasta files of our sequence (ITS2 region) were added as supplementary information (please see attachment).

- Figure 3: I would think it would be more appropriate to identify the area with yellow or other colored circles.

Arrows were replaced by yellow circles.

- The words that make up the phrase In vitro should be in italics.

It was italicized throughout the text.

- Table 1. I would suggest that the references go in the last column and it is not necessary to include the author's name or the year, since the reference number is more than enough.

Table 1 was modified according to reviewer’s suggestion.

We applied the same change to Table 2 to better harmonize the two tables.

- References must be prepared in accordance with MDPI's specifications

All references were rechecked and adapted to MDPI authors’ instructions.

-  Also make all changes requested in change control within the paper

All changes requested were made in change control mode.
